# Vitamin D3 Supplementation in Overweight/Obese Pregnant Women: No Effects on the Maternal or Fetal Lipid Profile and Body Fat Distribution—A Secondary Analysis of the Multicentric, Randomized, Controlled Vitamin D and Lifestyle for Gestational Diabetes Prevention Trial (DALI)

**DOI:** 10.3390/nu14183781

**Published:** 2022-09-14

**Authors:** Jürgen Harreiter, Lilian C. Mendoza, David Simmons, Gernot Desoye, Roland Devlieger, Sander Galjaard, Peter Damm, Elisabeth R. Mathiesen, Dorte M. Jensen, Lise Lotte T. Andersen, Fidelma Dunne, Annunziata Lapolla, Maria G. Dalfra, Alessandra Bertolotto, Ewa Wender-Ozegowska, Agnieszka Zawiejska, David Hill, Judith G. M. Jelsma, Frank J. Snoek, Christof Worda, Dagmar Bancher-Todesca, Mireille N. M. van Poppel, Rosa Corcoy, Alexandra Kautzky-Willer

**Affiliations:** 1Gender Medicine Unit, Division of Endocrinology and Metabolism, Department of Medicine III, Medical University of Vienna, 1090 Vienna, Austria; 2Institut d’Investigació Biomèdica Sant Pau (IIB SANT PAU), 08025 Barcelona, Spain; 3Macarthur Clinical School, Western Sydney University, Sydney 2560, Australia; 4Department of Obstetrics and Gynecology, Medical University of Graz, 8036 Graz, Austria; 5Department of Development and Regeneration, KU Leuven, University Leuven, 3000 Leuven, Belgium; 6Department of Obstetrics and Gynecology, University Hospitals Leuven, 3000 Leuven, Belgium; 7Department of Obstetrics, Gynecology and Fertility, GZA Sint-Augustinus, 2610 Wilrijk, Belgium; 8Department of Obstetrics and Gynaecology, Division of Obstetrics and Prenatal Medicine, Erasmus MC, University Medical Centre Rotterdam, 3015 GD Rotterdam, The Netherlands; 9Center for Pregnant Women with Diabetes, Departments of Endocrinology and Obstetrics, Rigshospitalet, 2100 Copenhagen, Denmark; 10Department of Clinical Medicine, Faculty of Health and Medical Sciences, University of Copenhagen, 1165 Copenhagen, Denmark; 11Steno Diabetes Center Odense, Odense University Hospital, 5000 Odense, Denmark; 12Department of Gynaecology and Obstetrics, Odense University Hospital, 5000 Odense, Denmark; 13Department of Clinical Research, Faculty of Health, University of Southern Denmark, 5000 Odense, Denmark; 14Clinical Research Facility (CRF) and National University of Ireland, H91 TK33 Galway, Ireland; 15Department of Medicine, Universita Degli Studi di Padova, 35128 Padova, Italy; 16Azienda Ospedaliero-Universitaria Pisana , 56126 Pisa, Italy; 17Department of Reproduction, Medical Faculty I, Poznan University of Medical Sciences, 60-525 Poznan, Poland; 18Lawson Health Research Institute, London, ON N6A 4V2, Canada; 19Amsterdam UMC, Department of Public and Occupational Health, Amsterdam Public Health Research Institute, Vrije Universiteit Amsterdam, 1081 BT Amsterdam, The Netherlands; 20Amsterdam UMC, Department of Medical Psychology, Amsterdam Public Health Research Institute, Vrije Universiteit Amsterdam, 1007 MB Amsterdam, The Netherlands; 21Division of Obstetrics and Feto-Maternal Medicine, Department of Obstetrics and Gynecology, Medical University of Vienna, 1090 Vienna, Austria; 22Institute of Human Movement Science, Sport and Health, University of Graz, 8010 Graz, Austria; 23CIBER Bioengineering, Biomaterials and Nanotechnology, Instituto de Salud Carlos III, 28029 Madrid, Spain; 24Gender Institute, La Pura, 3571 Gars am Kamp, Austria

**Keywords:** vitamin D, lipids, triglycerides, cholesterol, free fatty acids, overweight, obesity pregnancy, pregnancy outcomes, birth outcomes, cord blood, skinfolds, body fat distribution

## Abstract

Vitamin D deficiency is a common finding in overweight/obese pregnant women and is associated with increased risk for adverse pregnancy outcome. Both maternal vitamin D deficiency and maternal obesity contribute to metabolic derangements in pregnancy. We aimed to assess the effects of vitamin D3 supplementation in pregnancy versus placebo on maternal and fetal lipids. Main inclusion criteria were: women <20 weeks’ gestation, BMI ≥ 29 kg/m^2^. Eligible women (*n* = 154) were randomized to receive vitamin D3 (1600 IU/day) or placebo. Assessments were performed <20, 24–28 and 35–37 weeks and at birth. Linear regression models were used to assess effects of vitamin D on maternal and cord blood lipids. In the vitamin D group significantly higher total 25-OHD and 25-OHD3 levels were found in maternal and cord blood compared with placebo. Adjusted regression models did not reveal any differences in triglycerides, LDL-C, HDL-C, free fatty acids, ketone bodies or leptin between groups. Neonatal sum of skinfolds was comparable between the two groups, but correlated positively with cord blood 25-OH-D3 (r = 0.34, *p* = 0.012). Vitamin D supplementation in pregnancy increases maternal and cord blood vitamin D significantly resulting in high rates of vitamin D sufficiency. Maternal and cord blood lipid parameters were unaffected by Vitamin D3 supplementation.

## 1. Introduction

Vitamin D is an important fat-soluble vitamin with steroid hormone function. It is predominately known for its essential role in calcium homeostasis and bone mineralization [1]. Vitamin D receptors can be found in several organs and tissues throughout the body indicating the pleiotropic role of vitamin D with involvement in metabolism, inflammatory response, oxidative stress, modulation of cell growth, neuromuscular function and immunity [1]. Vitamin D deficiency is a common pregnancy finding across Europe and is associated with increased risk for adverse pregnancy and birth outcome with higher risks for gestational diabetes mellitus, dyslipidemia, preeclampsia, preterm birth, low birth weight, higher infection rates and even disturbed brain development or respiratory function [2,3]. Higher rates of vitamin D deficiency in pregnancy are observed in obese pregnant women [4]. A two-fold higher risk for maternal and also for neonatal vitamin D deficiency, which increased with higher BMI, has been reported [4]. Other studies have corroborated this finding of lower vitamin D levels in offspring of obese women and confirmed the relationship of maternal obesity with the nutritional status of the offspring [5]. Both maternal vitamin D deficiency and maternal obesity contribute to maternal metabolic derangements in pregnancy including hyperglycemia and dyslipidemia [6,7,8]. Prior studies have found inconclusive results with some studies demonstrating positive effects of vitamin D supplementation on lipid levels in pregnancies. Most of these studies were based on an observational or retrospective designs, were performed mainly in women with gestational diabetes, were started with the intervention late in pregnancy, had low participant numbers and short follow up time, had low or very high Vitamin D dose substitution, had no fetal follow up or no control groups for comparison [8,9,10,11,12]. A systematic review with six RCTs found significant improvements in insulin resistance, beta-cell function and LDL cholesterol, but no changes in triglycerides, HDL cholesterol, fasting glucose or insulin levels [8]. Another trial supplementing 50,000 IU vitamin D3 every 2 weeks for 8 weeks in women with GDM and starting in 24–28 weeks of gestation found lower fasting glucose and HbA1c, but no significant differences in lipid parameters [12]. In pregnancy, lipids increase physiologically due to pregnancy-related hormonal changes. Nonetheless, dyslipidemia in pregnancy is an important risk factor associated with adverse pregnancy outcomes and adverse fetal development [13,14]. Little information is available from randomized controlled studies about the effects of vitamin D supplementation on lipid metabolism throughout pregnancy in obese, pregnant women not affected by GDM. Thus, further clinical investigations on vitamin D supplementation starting early in pregnancy with assessment of maternal parameters throughout pregnancy and fetal and pregnancy outcomes are of clinical significance. Given the previously reported positive effects of vitamin D supplementation on metabolic parameters, including lipids, in pregnant women with GDM and based on observational studies demonstrating inverse associations of maternal vitamin D levels with adverse pregnancy and birth outcomes [11,15], we hypothesized that beneficial effects of vitamin D supplementation on lipid metabolism in overweight/obese pregnant women would be expected.

In this secondary analysis of the DALI Vitamin D study, including overweight/obese pregnant women with normal glucose tolerance at randomization before 20 weeks’ gestation, we aimed to assess the effects of a vitamin D supplementation with 1600 IU Vitamin D3 per day starting in early pregnancy versus placebo on maternal and fetal lipid parameters, body fat distribution as well as pregnancy outcomes.

## 2. Materials and Methods

### 2.1. Study Design and Participants

The vitamin D And LIfestyle for gestational diabetes prevention trial (DALI) was a prospective European-wide multicentre randomized controlled trial (RCT) designed to investigate the potential preventive effects of different lifestyle approaches or vitamin D3 supplementation on GDM progression in obese pregnant women recruited prior to 20 weeks gestation, as previously reported elsewhere [16,17,18,19]. The Vitamin D intervention trial included women (*n* = 154) from 7 European countries (United Kingdom, Ireland, Austria, Poland, Italy (2 centres), Spain and Belgium) aged ≥ 18 years, singleton pregnancy before 20 weeks of gestation, pre-pregnancy BMI ≥ 29 kg/m^2^, and ability to give informed consent. Exclusion criteria included pre-existing diabetes, need for complex diets, inability to walk ≥ 100 m safely, significant chronic medical conditions, current or past abnormal calcium metabolism (hypo/hyperparathyroidism, nephrolithiasis, hypercalciuria or hypercalcemia). Recruitment was conducted between 2012 and 2015. The DALI trial was approved by each local research ethics committee. Written informed consent was obtained from all participating subjects. DALI was registered in ISRCTN registry (ISRCTN70595832) and funded by the European Union 7th framework program (Grant Agreement no. 242187).

### 2.2. Study Procedures and Study Intervention

Randomization procedures, blinding and sample size calculations were described elsewhere [16,19]. Briefly, obese pregnant women were randomized after a baseline examination to receive either lifestyle advice (healthy eating and physical activity) and placebo, lifestyle advice and vitamin D supplementation, placebo alone, or vitamin D alone using a computerized electronic random number generator, pre-stratified for site. As in the primary outcome trial no interaction between the lifestyle and vitamin D intervention was observed [19], groups randomized to vitamin D supplementation and groups randomized to placebo were combined for further analysis. At 24–28, 35–37 weeks of gestation and at delivery, follow up visits were performed. The vitamin D study aimed to achieve maternal vitamin D concentrations ≥ 50 nmol/L at term. The determination of vitamin D cut off levels was described in a previous manuscript [19].

Pregnant women were randomized to either receive 1600 IU vitamin D3 or placebo per day. The intervention was administered orally. Devaron^®^ tablets with 400 IU of vitamin D3 were produced by Vemedia (Diemen, Netherlands) (RVG 09766). Placebo tablets, identical to the Devaron^®^ tablets in appearance, were produced by Vemedia especially for the DALI trial. The study was designed to achieve an average total vitamin D intake of 2000 IU/day in the vitamin D arm accounting for an additional 400 IU/day of vitamin D from pregnancy multivitamin compounds to the 1600 IU/day from the intervention, since the use of multivitamin compounds during pregnancy is common. Our decision to choose 2000 IU/day vitamin D3 was based on a vitamin D supplementation study, which observed safe application and vitamin D sufficiency in pregnant women with daily doses at 2000 IU/day [20].

### 2.3. Assessments

Assessments and blood sampling were performed before 20 weeks, between 24–28 weeks and 35–37 weeks. Venous cord blood was drawn immediately after birth. GDM was diagnosed according to the International Association of the Diabetes and Pregnancy Study Groups/ World Health Organization (IADPSG/ WHO) 2013 GDM criteria and those with GDM at baseline were excluded from participation in the intervention trial. Thresholds for hypercalciuria and hypercalcemia were reported previously [19]. We obtained individual and medical information through questionnaires including demographics, pre-pregnancy weight, maternal smoking, past/current medical obstetric and medication history and current medication and/or multivitamin intake. At the study visits information on the month of blood collection, and the number of vitamin D/placebo pills missed was collected. The measurements of height, weight, neck circumferences, skinfolds and blood pressure were explained in detail in previous publications [16,17]. Maternal skinfolds at four different sites (biceps, triceps, subscapular, supra-iliac) were measured at baseline and 35–37 weeks. Shortly after birth neonatal skinfolds were measured.

### 2.4. Behaviour Assessment

We used self-reporting tools for the assessment of physical activity and dietary behavior [18]. Physical activity was assessed by the Pregnancy Physical Activity Questionnaire, which gives estimates of time spent on activity and sedentary behavior [21]. Based on prior work, food intake was assessed with a 12-item questionnaire asking self-reported frequencies (i.e., days per week) and amounts (i.e., portions per day) of consumed food [22]. The calculations of proxies of nutritional components (i.e., fiber, protein, carbohydrate, fat), portion size and sugar drinks were reported elsewhere [18]. All values are given in total number of items consumed per week.

### 2.5. Laboratory Analyses

Blood samples were obtained fasting. Blood samples were centrifuged, and aliquots were stored in an ISO 9001 certified central laboratory (Graz, Austria) at −20 °C or −80 °C until further analysis. Serum 25(OH)D concentrations were assessed using a ClinMass^®^ liquid chromatography mass spectrometry/ mass spectrometry (LCeMS/MS) complete kit (RECIPE Chemicals + Instruments GmbH, Munich, Germany). The kit includes a solid phase extraction buffer, mobile phase, autosampler washing solution, precipitation reagent, serum calibrators and quality control samples. Moreover, RECIPE provided the HPLC column, solid phase extraction column and an inline filter. The precipitation reagent, which was added to serum, contained D6-25(OH) D3 as internal standard. The samples were vortexed, centrifuged and subsequently 20 mL of supernatant were injected into an Advance UHPLC in online extraction mode coupled to an EVOQ Elite (Bruker Daltonics, Bremen, Germany) triple quadrupole mass spectrometer running at the following mass transitions: 25(OH)D3 383.2/257.2/211.2; D6-25-hydroxyvitamin D3 389.2/263.2/211.2; 25(OH)D2 395.2/209.2/269.1. Calibration curves including the internal standard were used to perform data processing. Estimates of chromatographically separated epi-25(OH) D3 were used for further analysis due to a missing internal standard and calibration curve External validation of the analysis was performed comparing samples with a Vitamin D External Quality Assessment Scheme (DEQAS) certified center (kindly enabled by Prof. Kevin Cashman and Ms. Kirsten Dowling, Cork Centre for Vitamin D and Nutrition Research, University College Cork) demonstrating valid results as reported previously [19].

Analytical procedures for metabolic parameters (triglycerides (TG), total cholesterol (TC), HDL cholesterol (HDL-C), LDL cholesterol (LDL-C), non-esterified (FFA), 3-hydroxy butyrate (3BHB), leptin, glucose and insulin) were reported previously [7,17,18].

### 2.6. Statistical Analysis

Continuous variables were summarized by means and standard deviations and categorical variables by counts and percentages. Assumption of Gaussian distribution of parameters was decided by visual assessment of histograms and calculation of skewness and kurtosis. Skewed data were log transformed before further analysis. Comparison for continuous variables between intervention groups were performed using *t*-test and using Chi^2^ test for binary data. Randomized women were compared with excluded women using *t*-test. Pearson correlation coefficients were used for correlation analysis. Comparisons of triglyceride, LDL-C, HDL-C, FFA, 3BHB and leptin from baseline to 24–28 and 35–37 weeks of pregnancy were made between groups using general linear regression models. Models were adjusted either for baseline value of the outcome variable only or, in the fully adjusted model, for baseline value of the outcome variable, maternal age, BMI at baseline, weight gain until time of examination, gestational week, educational level, insulin resistance at time of examination (HOMA Index), self-reported food intake (portion size), self-reported physical activity, maternal smoking and study site. Models with weight gain variables were adjusted either for maternal BMI at baseline only and for maternal age, BMI at baseline, gestational week, educational level, insulin resistance at time of examination (HOMA Index), self-reported food intake (portion size), self-reported physical activity, maternal smoking and study site in the fully adjusted model. Dietary components and physical activity were analyzed after adjustment for baseline levels using general linear models.

Cord blood lipids were analyzed using general linear models, adjusting for gestational age at birth and fetal sex in a simple model and, in a fully adjusted model, for gestational age, fetal sex, maternal age, maternal BMI at baseline, maternal HOMA Index at baseline, weight gain in pregnancy, study site and maternal baseline outcome variable if applicable. Statistical analysis was performed using SPSS 27.0 (SPSS Inc., Chicago, IL, USA) and GraphPad Prism 9.3.0 (GraphPad Software, La Jolla, CA, USA). A two-sided *p*-value < 0.05 was considered statistically significant. Data were analyzed according to the intention-to-treat principle, blinded for group allocation.

## 3. Results

### 3.1. Baseline Characteristics

In total, 213 obese pregnant women were included and assessed for screening (Figure 1). Of these 154 obese pregnant women were randomized to receive either Vitamin D or placebo.

In the trial population European ancestry and LDL—cholesterol levels at baseline—were significantly higher in the vitamin D group compared with the placebo group, while the other assessed parameters at baseline were comparable between the two treatment arms. Pregnancy vitamins were used in a high percentage of participants in both study groups (Table 1) and baseline vitamin D levels were not significantly different between groups. Vitamin D sufficiency at baseline was reported in 55/73 (75%) in placebo and 66/78 (85%, *p* = n.s.) in women of the vitamin D group.

Patients excluded from participation (*n* = 33) were not significantly different in most of the assessed baseline parameters compared with randomized women (Table 1). Only the skinfold measurement at the biceps was significantly lower. Lipid parameters as well as vitamin D parameters at baseline were comparable. Baseline characteristics are summarized in Table 1.

### 3.2. Intervention Effects on Maternal Vitamin D and Lipid Levels

Maternal lipid parameters and vitamin D levels as well as anthropometric data throughout the pregnancies are shown in Table 2. Significant differences were found between the vitamin D supplementation group and placebo group in total 25-OH-D and 25-OH-D3 but not 25-OH-D2 at 24–28 and 35–38 weeks of gestation. Vitamin D sufficiency was high in both groups, but significantly higher in the vitamin D supplementation group compared with placebo at both time points (74/76 (97%) vs. 49/66 (74%) and 63/64 (98%) vs. 43/55 (78%), *p* < 0.001 both). Calcium levels or albumin corrected calcium were not different between vitamin D or placebo groups throughout pregnancy.

LDL-C levels continued to be significantly higher in the vitamin D group at 24–28 weeks of gestation and were close to significant higher levels at 35–37 weeks of gestation. However, after correction for baseline parameters or after full adjustment no significant differences in LDL-C or triglycerides, HDL-C, free fatty acids, ketone bodies and leptin at 24–28 and 35–37 weeks of gestation were observed, while in the vitamin D group significant higher 25OH D3 and total 25OH D levels were detected at both time points compared with placebo (Table 2). Adjustment for ethnicity did not change maternal lipid parameters or skinfolds significantly (data not shown).

Nutritional behavior and physical activity throughout pregnancy are shown in Table 3. Self-reported nutritional habits and physical activity did not differ between groups throughout pregnancy.

### 3.3. Cord Blood Lipid and Vitamin D Levels and Birth Outcomes

Pregnancy and birth outcomes did not reveal any significant differences between the vitamin D and placebo group (Table 4). Cord blood lipid parameters, ketone bodies and leptin were not significantly different, which was also seen after full adjustment. Adjustment for ethnicity did not change birth outcome, fetal lipid parameters or skinfolds significantly (data not shown). 25-OH-D3 and total 25-OH-D levels in cord blood were significantly higher in the vitamin D intervention group compared with placebo. Vitamin D sufficiency with 25-OH-D levels above 50 nmol/L assessed in cord blood was significantly lower in placebo (*n* = 16/25, 64%) compared with the vitamin D group (*n* = 30/34, 88%, *p* < 0.05).

### 3.4. Correlation Analysis

Correlation analysis revealed no correlations between change in maternal 25-OH-D3 levels throughout pregnancy and triglycerides (r = 0.12, *p*= 0.26), HDL-C (r = −0.01, *p* = 0.89), LDL-C (r = 0.06, *p* = 0.55), FFA (r = 0.06, *p* = 0.58) or 3BHB (r = −0.07, *p* = 0.50) at weeks 35–37. Moreover, change in maternal 25-OH-D3 levels was not correlated with weight gain in pregnancy (r = −0.13, *p* = 0.11), maternal sum of skinfolds (r = 0.02, *p* = 0.84) or maternal leptin levels at 35–37 weeks of gestation (r = −0.04, *p* = 0.67).

Changes of maternal 25-OH-D3 levels were significantly correlated with cord blood 25-OH-D3 levels (r = 0.68, *p* < 0.001). However, no correlations were found between maternal 25-OH-D3 levels and gestational age at delivery, birth weight, neonatal skinfolds or cord blood lipid parameters, ketone bodies or leptin. In cord blood, 25-OH-D3 levels were not correlated with any lipid parameters. A positive correlation was found for neonatal sum of skinfolds and 25-OH-D3 (r = 0.34, *p* = 0.012) in cord blood. However, cord blood leptin did not show any correlation with vitamin D.

## 4. Discussion

In our analysis we found a significant increase of maternal vitamin D levels throughout pregnancy and also in cord blood in the supplementation group compared with the placebo group, but did not find any significant differences between the treatment arms in maternal lipid parameters nor in fetal lipid parameters or birth outcomes. In contrast to existing vitamin D supplementation studies in pregnancy, mostly assessing efficacy and safety of vitamin D application or effects on bone metabolism, the novelty of the DALI study is the predefined design with the aim to test the effects of lifestyle or vitamin D intervention in the prevention of GDM and metabolic parameters such as lipids in overweight/obese women in a multicentre, randomized controlled trial. Moreover, we examined vitamin D supplementation from before 20 weeks of pregnancy to end of pregnancy and evaluated the effects on fetal cord blood and pregnancy and birth outcomes.

### 4.1. Maternal Outcomes

The effects of vitamin D supplementation or associations of vitamin D and lipid metabolism in pregnancy were analyzed in several previous studies with diverging results. In our analysis we found physiologically increasing lipid parameters throughout pregnancy in both groups as reported in earlier studies [23,24]. Supplementation of vitamin D3 did not result in any differences in triglycerides, LDL-C, HDL-C, FFA, 3BHB or leptin compared with the placebo group at 24–28 and 35–37 weeks of gestation. A recently published RCT assessing the efficacy of two different vitamin D3 doses in pregnancy (1000 vs. 2000 IU/day), but without a placebo group, found significant and safe increases of vitamin D from first trimester to week 34 in both treatment groups, which is comparable to our results [25]. The authors observed that throughout pregnancy, lipid parameters (triglycerides, total cholesterol, HDL-C, LDL-C) increased significantly in both treatment arms, but between group differences in third trimester were not significantly different. Cord blood vitamin D >=50 nmo/l was achieved in the majority of participants in both treatment arms, which is in line with our findings, but differently to our investigation fetal, birth outcomes or cord blood lipid parameters were not reported. A further RCT in women with GDM comparing supplementation with 50,000IU vitamin D3 for 8 weeks with placebo found no differences in lipid parameters between groups [12]. Lower triacylglycerol, total cholesterol and LDL cholesterol levels were found in the vitamin D group of a further trial investigating the effects of a 16- week vitamin D supplementation with fortified yoghurt starting in the second trimester in women with GDM [26]. Divergent results were reported in observational studies. Significantly higher triglyceride, HDL and LDL cholesterol levels were found in women with GDM with higher vitamin D levels compared with lower vitamin D levels [11]. In a population with high insufficiency/ deficiency rates, weak but significant positive associations of first trimester vitamin D levels with total cholesterol, triglycerides and corrected calcium were reported [27].

### 4.2. Birth Outcomes

Associations between fetal and neonatal growth, birth outcomes and vitamin D levels in pregnancy were reported in several studies but with conflicting results [10,15]. In our study we identified no differences in gestational age at delivery, birth weight, height, skinfolds, head or abdominal circumference between the treatment groups. Similar to our results, other studies found no differences in maternal and fetal pregnancy outcomes, as gestational week of birth, birth weight and body length [11]. Furthermore, RCTs investigating high dose vitamin D3 supplementation up to 4000 IU per day found no differences in birth or pregnancy outcomes [20]. However, recent studies show strong associations of vitamin D deficiency with fetal growth retardation in early pregnancy, which was further increased in overweight/obese women [10] and third-trimester fetal growth restriction [15]. Women with low Vitamin D levels also had higher risk for preterm birth and SGA births [15], which was corroborated by a computational model for the early prediction of SGA [28]. Divergent results with increases in neonatal body weight, length and head circumferences as well as higher maternal weight gain in pregnancy were found in another RCT including pregnant women with vitamin D deficiency [9]. A meta-analysis of RCTs found higher birth weight and length in the vitamin D supplementation group [29].

In our study we found a significant correlation between cord blood, but not change in maternal, vitamin D3 levels and neonatal sum of skinfolds, which might also be an accidental finding (type 1 error). Interestingly, cord blood leptin was also not correlated with cord blood or maternal vitamin D. An older study reported that lower maternal vitamin D levels assessed at week 34 of gestation were associated with lower fetal fat mass at birth but with higher fat mass at ages 4 and 6 years [30]. The authors speculate that maternal vitamin D deficiency seems to be a relevant factor in fetal development and genesis of adiposity. This was corroborated by recent studies, which observed significantly higher fetal fat mass measured with DXA in offspring of obese vitamin D deficient mothers [31] and higher abdominal subcutaneous adipose tissue volume in magnetic resonance imaging (MRI) also after accounting for maternal glucose [32]. Recent studies discussed higher vitamin D levels to be a proxy for healthier lifestyle including outdoor physical activity and healthier food intake and positive associations of exercise and vitamin D levels were reported [31]. Finally, a healthier maternal lifestyle in pregnancy influences gene expression involved in fat accumulation, body composition and metabolism and might act positively on the competitive intrauterine milieu transferring more maternal resources to maternal muscle and reducing energy substrates available to fetal adipocytes [31,33].

A rat model investigating the effects of vitamin D deficiency in offspring found that vitamin D-deficient offspring presented with higher blood lipid levels, blood glucose and adipose tissue [34]. A DNA methylation analysis found that differences with the control group were found in more than 300 involved genes. Due to this methylation alterations and differences in preadipocyte proliferation of preadipocytes, the authors speculated a tight association of vitamin D deficiency and offspring obesity in their rat models.

### 4.3. Limitations and Strengths

In our analysis we found a significant increase of vitamin D from baseline to week 35–37 of gestation and observed that more than 98% of the participating women in the supplementation group had sufficient vitamin D levels at or above 50 nmol/l [19]. However, the placebo group also had surprisingly high levels of vitamin D sufficiency already at baseline and more than 75% of the placebo group had sufficient vitamin D levels at the end of pregnancy. Interestingly a previous review with focus on the effects of vitamin D supplementation on inflammation mentioned that studies with insufficient/deficient baseline 25-OH-D levels had higher chances to report beneficial effects compared with those studies with sufficient 25-OH-D levels [35]. Moreover, many women, and especially those participating in a RCT are willing to change their habits in pregnancy. The usual care group might also have taken additional supplements or pregnancy vitamins weakening our chance of finding differences in metabolic parameters substantially. However, our records regarding pregnancy vitamin intake in pregnancy did not change significantly from baseline and was not significantly different between groups. The season of blood examination was equally distributed over the study groups and food intake patterns were not different between groups. There was a substantial difference in distribution of ethnicity between vitamin D and placebo groups, which could potentially influence outcomes significantly, but additional adjustment for ethnicity did not change the results.

Questionnaires assessing food intake and physical activity relied on self-reporting. Under or over reporting due to social desirability and recall bias are relevant factors, which need to be considered. Moreover, calculation of detailed information of macronutrient content is not possible with our food questionnaire. We cannot extrapolate results from our trial to women with lower BMIs, as these populations were not included in our trial and might serve as an explanation for controversial results. Moreover, we were not able to reach the intended sample size.

A strength of this study is the inclusion of a homogenous population of overweight/obese women with normal glucose tolerance at baseline. Different to many studies the vitamin D levels were followed throughout pregnancy starting before 20 weeks of gestation until birth and cord blood was collected.

## 5. Conclusions

In conclusion, vitamin D supplementation with 1600 IU/day vitamin D3 in overweight/obese pregnant women led to a safe and significant increase of vitamin D levels in pregnancy and high prevalence of vitamin D sufficiency without safety issues. However, we were not able to detect any differences in maternal or neonatal lipid parameters in comparison with the placebo group. From this point of view daily vitamin D3 supplementation in overweight/obese pregnant women with high levels of vitamin D sufficiency does not improve maternal or fetal lipid parameters, body fat distribution or pregnancy outcomes and should not be used in pregnancy for these indications. Due to controversial results in the literature further studies are necessary to elucidate the role of vitamin D on lipid metabolism in pregnancy and offspring adiposity.

## Figures and Tables

**Figure 1 nutrients-14-03781-f001:**
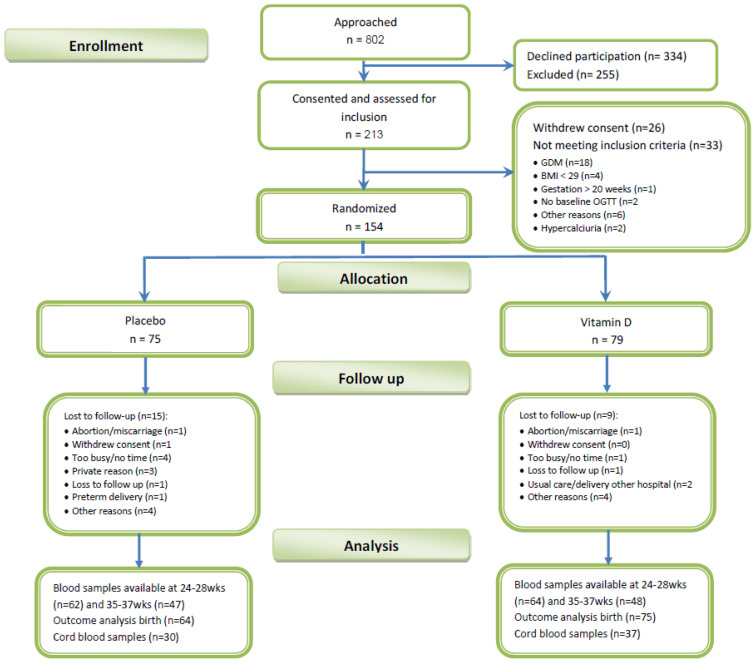
Study flow chart.

**Table 1 nutrients-14-03781-t001:** Baseline characteristics of women participating in the DALI Vitamin D trial.

	Vitamin D N = 79	Placebo N = 75	*p*	Total N = 154	Excluded N = 33	*p*
	Mean (SD)	Mean (SD)		Mean (SD)	Mean (SD)	
Age (year)	32.8 (5.4)	32.2 (5.2)	ns	32.5 (5.3)	33.5 (4.6)	ns
Height (cm)	165.1 (6.6)	164.7 (7.4)	ns	164.9 (7.0)	163.2 (4.9)	ns
Pre-pregnancy weight (kg)	92.0 (13.8)	90.3 (13.4)	ns	91.2 (13.7)	91.4 (16.1)	ns
Pre pregnancy BMI (kg/m^2^)	33.7 (4.3)	33.3 (4.3)	ns	33.5 (4.3)	34.3 (6.0)	ns
Weight gain until 1st visit (kg)	2.3 (4.8)	1.3 (3.4)	ns	1.8 (4.2)	2.4 (3.5)	ns
Gestational week at enrollment (weeks)	15.0 (2.9)	15.4 (2.5)	ns	15.2 (2.7)	15.3 (2.8)	ns
	N (%)	N (%)		N (%)	N (%)	
European descent	72/79 (91.1)	59/75 (78.7%)	<0.05	131/154 (85.1)	30/33 (90.9)	ns
Higher education	50/78 (64.1)	45/75 (60.0)	ns	95/153 (62.1)	18/33 (54.5)	ns
Multiparity	48/79 (60.8)	46/75 (61.3)	ns	94/154 (61.0)	21/33 (63.6)	ns
Diabetes in family	12/79 (15.2)	17/75 (22.7)	ns	29/154 (18.8)	11/32 (34.4)	ns
History of GDM	7/46 (15.2)	7/46 (15.2)	ns	14/92 (15.2)	4/19 (21.1)	ns
Previous unexplained stillbirth	5/45 (13.3)	3/46 (6.5)	ns	9/91 (9.9)	2/19 (10.5)	ns
History of malformation	4/47 (8.5)	1/45 (2.2)	ns	5/92 (5.4)	1/19 (5.2)	ns
Prev. macrosomia	15/47 (31.9)	11/46 (23.9)	ns	26/93 (28.0)	6/19 (31.6)	ns
PCOS	10/79 (12.7)	11/74 (14.9)	ns	21/153 (13.7)	6/33 (18.2)	ns
Chronic hypertension	14/79 (17.7)	9/74 (12.2)	ns	23/153 (15.0)	7/33 (21.1)	ns
Smoking	17/79 (22.5)	14/75 (18.7	ns	31/154 (20.1)	7/33 (21.2)	ns
Alcohol in pregnancy	6/76 (7.9	5/74 (6.8)	ns	11/150 (7.3)	2/33 (6.1)	ns
Taking pregnancy vitamins	68/79 (86.1%)	63/75 (84.0%)	ns	131/154 (85.1)	32/33 (97.0)	ns
	Mean (SD)	Mean (SD)		Mean (SD)	Mean (SD)	
BP sys (mmHg)	115.6 (11.6)	114.9 (12.0)	ns	115.2 (11.7)	118.9 (11.2)	ns
BP dia (mmHg)	72.4 (9.3)	71.4 (9.2)	ns	71.9 (9.2)	75.0 (8.6)	ns
Heart rate (bpm)	79.2 (10.3)	79.1 (9.7)	ns	79.2 (10.0)	82.4 (11.7)	ns
Triglycerides (mmol/L)	1.32 (0.41)	1.41 (0.55)	ns	1.36 (0.48)	1.40 (0.64)	ns
LDL-C (mmol/L)	3.21 (0.72)	2.96 (0.73)	<0.05	3.08 (0.74)	3.04 (0.70)	ns
HDL-C (mmol/L)	1.37 (0.23)	1.33 (0.28)	ns	1.35 (0.26)	1.40 (0.22)	ns
FFA (mmol/L)	0.62 (0.23)	0.64 (0.29)	ns	0.64 (0.26)	0.68 (0.22)	ns
3BHB (mmol/L)	0.067 (0.085)	0.057 (0.074)	ns	0.062 (0.080)	0.087 (0.096)	ns
Leptin (ng/dL)	34.5 (17.6)	33.7 (17.6)	ns	34.0 (17.1)	36.9 (20.3)	ns
Calcium (mmol/L)	2.2 (0.3)	2.2 (0.4)	ns	2.2 (0.3)	2.3 (0.1)	ns
Albumin (g/L)	36.5 (3.6)	36.1 (3.9)	ns	36.3 (3.8)	37.0 (4.5)	ns
Albumin-corrected Calcium (mmol/L)	2.2 (0.3)	2.2 (0.4)	ns	2.2 (0.4	2.3 (0.1)	ns
25OH D2 (nmol/L)	4.2 (4.6)	3.0 (4.3)	ns	3.6 (4.5)	3.6 (4.6)	ns
25OH D3 (nmol/L)	69.3 (26.8)	66.7 (26.8)	ns	68.0 (26.7)	63.3 (25.4)	ns
25OH D total (nmol/L)	73.40 (26.8)	69.8 (26.9)	ns	71.6 (26.8)	66.9 (25.9)	ns
Skinfold biceps (mm)	20.1 (7.9)	19.8 (6.4)	ns	20.0 (7.2)	17.1 (7.2)	<0.05
Skinfold triceps (mm)	28.2 (6.6)	28.1 (6.0)	ns	28.1 (6.3)	26.9 (5.7)	ns
Skinfold subscapular (mm)	28.6 (7.4)	30.3 (9.4)	ns	29.4 (8.4)	30.5 (6.8)	ns
Skinfold supra-iliac (mm)	34.3 (7.8)	33.3 (8.8)	ns	33.8 (8.3)	31.8 (7.1)	ns
Sum of skinfolds (mm)	111.3 (20.5)	111.5 (24.2)	ns	111.4 (22.3)	106.3 (19.6)	ns
Fat percentage (%)	31.1 (2.9)	31.2 (3.2)	ns	31.1 (3.0)	30.8 (2.3)	ns
Neck circumference (cm)	36.3 (2.3)	36.4 (2.0)	ns	36.4 (2.2)	36.6 (2.6)	ns

SD = standard deviation, *p* = level of significance, ns = not significant, BMI = body mass index, GDM = gestational diabetes mellitus, PCOS = polycystic ovary syndrome, BP = blood pressure, sys = systolic, dia = diastolic, bpm = beats per minute, LDL-C = Low Density Lipoprotein cholesterol, HDL-C = High Density Lipoprotein cholesterol, FFA = free fatty acid, 3BHB = beta-hydroxy butyrate, 25OH D = 25 hydroxyvitamin.

**Table 2 nutrients-14-03781-t002:** Primary and secondary outcomes in the two intervention groups and intervention effects at 24–28 and 35–37 weeks of pregnancy.

	Vitamin D *n* = 77		Placebo *n* = 67		Vit.D vs. Placebo $	Vit.D vs. Placebo $$
24–28 weeks	Mean	SD	Mean	SD	mean difference (95% CI)	Adjusted mean difference (95% CI)
Gestational age, weeks	26.5	1.4	26.3	1.6	0.3 (−0.2; 0.8)	0.3 (−0.2; 0.8)
Triglyceride, mmol/L	1.86	0.56	1.82	0.59	0.08 (−0.08; 0.24)	0.05 (−0.13; 0.22)
LDL-C, mmol/L	3.99 *	0.96	3.55 *	0.83	0.20 (−0.03; 0.43)	0.20 (−0.05; 0.44)
HDL-C, mmol/L	1.47	0.21	1.45	0.28	0.00 (−0.05; 0.05)	0.01 (−0.05; 0.06)
FFA, mmol/L	0.54	0.22	0.52	0.17	0.014 (−0.047; 0.076)	0.015 (−0.053; 0.083)
3BHB, mmol/L	0.087	0.097	0.053	0.084	0.018 (−0.012; 0.048)	0.007 (−0.027; 0.040)
Leptin, ng/dL	33.0	17.5	33.0	15.3	−0.4 (−5.1; 4.3)	0.9 (−3.6; 5.4)
Calcium, mmol/L	2.1	0.3	2.1	0.4	0.02 (−0.07; 0.10)	−0.02 (−0.08; 0.05)
Albumin-corrected Calcium, mmol/L	2.2	0.4	2.2	0.5	0.04 (−0.08; 0.15)	−0.01 (−0.03; 0.05)
25-OH-D2, nmol/L	2.9	4.3	3.3	4.7	−0.6 (−2.0; 0.8)	−0.7 (−2.3; 0.9)
25-OH-D3, nmol/L	116.6 ***	35.2	78.7 ***	39.3	35.1 (23.3; 46.9) ***	34.3 (22.5; 46.2) ***
25-OH-D total, nmol/L	119.5 ***	35.5	82.0 ***	39.4	34.3 (22.4; 46.1) ***	33.4 (21.5; 45.2) ***
Neck circumference, cm	36.6	2.2	36.6	2.0	0.1 (−0.2; 0.5)	0.1 (−0.3; 0.4)
Weight gain, kg	4.1	3.1	3.9	2.6	0.2 (−0.7; 1.1)	−0.4 (−1.3; 0.6)
	Vitamin D *n* = 70		Placebo *n* = 59		Vit.D vs. Placebo $	Vit.D vs. Placebo $$
35–37 weeks	Mean	SD	Mean	SD	mean difference (95% CI)	Adjusted mean difference (95% CI)
Gestational age, wks	35.9	0.8	35.9	0.9	0.0 (−0.3; 0.3)	0.0 (−0.3; 0.3)
Triglyceride, mmol/L	2.49	0.73	2.29	0.80	0.19 (−0.08; 0.46)	0.11 (−0.18; 0.39)
LDL-C, mmol/L	4.06	1.01	3.68	0.93	0.17 (−0.14; 0.48)	0.17 (−0.14; 0.48)
HDL-C, mmol/L	1.41	0.25	1.44	0.30	−0.00 (−0.08; 0.07)	−0.00 (−0.09; 0.08)
FFA, mmol/L	0.54	0.19	0.54	0.20	−0.01 (−0.08; 0.07)	−0.04 (−0.12; 0.05)
3BHB, mmol/L	0.079	0.058	0.062	0.074	0.016 (−0.012; 0.043)	0.012 (−0.019; 0.042)
Leptin, ng/dL	32.7	14.3	31.7	13.0	0.38 (−4.6; 5.4)	1.2 (−3.9; 6.4)
Calcium, mmol/L	2.2	0.2	2.1	0.4	0.10 (−0.22; 0.41)	0.00 (−0.05; 0.05)
Albumin-corrected Calcium, mmol/L	2.3	0.3	2.2	0.4	0.04 (−0.08; 0.15)	−0.25 (−0.12; 0.07)
25-OH-D2, µg/L	2.3	4.1	2.9	4.5	−1.1 (−2.5; 0.3)	−1.3 (−2.8; 0.3)
25-OH-D3, µg/L	120.7 ***	38.9	81.7 ***	40.3	38.9 (24.6; 53.1) ***	36.2 (19.9; 52.5) ***
25-OH- D total, µg/L	123.0 ***	38.8	84.6 ***	39.8	37.9 (23.8; 51.9) ***	34.9 (18.8; 51.0) ***
Skinfold biceps, mm	21.0	9.0	20.8	8.0	−0.1 (−2.9; 2.9)	0.2 (−3.2; 3.5)
Skinfold triceps, mm	27.4	7.1	27.7	8.1	−0.4 (−2.7; 1.9)	−1.0 (−3.6; 1.7)
Skinfold subscapular, mm	29.6	8.4	29.7	10.2	0.8 (−2.0; 3.6)	−0.2 (−3.5; 3.2)
Skinfold supra-iliac, mm	34.7	13.6	33.7	11.7	−0.4 (−4.1; 3.3)	−3.3 (−7.6, 1.2)
Sum of skinfolds, cm	112.7	28.1	111.9	30.4	−0.5 (−9.0; 8.0)	−5.6 (−15.5; 4.3)
Fat percentage, %	31.3	3.0	31.3	3.9	0.0 (−1.0; 1.0)	−0.6 (−1.8; 0.6)
Neck circumference, cm	36.8	2.6	36.8	2.1	0.1 (−0.2; 0.5)	0.1 (−0.3; 0.4)
Weight gain, kg §	7.7	4.6	8.0	6.1	−0.3 (−2.1; 1.5)	−1.6 (−3.2; 0.1)

* *p* < 0.05, *** *p* < 0.001, SD = standard deviation, CI = confidence interval, LDL-C = Low Density Lipoprotein cholesterol, HDL-C = High Density Lipoprotein cholesterol, FFA = free fatty acid, 3BHB = beta-hydroxy butyrate, 25OH D = 25 hydroxyvitamin. $ adjusted for baseline values of the outcome variable, $$ adjusted for baseline values of the outcome variable, maternal age, BMI at baseline, weight gain until time of examination, gestational week, educational level, insulin resistance at time of examination (HOMA Index), self-reported food intake (portion size), self-reported physical activity, maternal smoking, study site. § Weight gain was adjusted for BMI at baseline in the unadjusted model and for maternal age, BMI at baseline, gestational week, educational level, insulin resistance at time of examination (HOMA Index), self-reported food intake (portion size), self-reported physical activity, maternal smoking, study site in the adjusted model.

**Table 3 nutrients-14-03781-t003:** Nutritional and physical activity information in the two intervention groups and mean differences between intervention groups adjusted for baseline levels.

	Vit D N = 77	Placebo N = 62	
Before 20 weeks of gestation	Mean (SD)	Mean (SD)	
Sugar drinks (n/week)	7.7 (12.0)	5.3 (7.1)	
Fiber (n/week)	29.6 (31.4)	25.6 (12.2)	
Protein (n/week)	8.8 (10.0)	8.5 (6.1)	
Fat (n/week)	6.6 (6.0)	5.9 (4.8)	
Carbohydrates (n/week)	39.0 (38.8)	32.9 (15.7)	
Portion size (n/week)	20.1 (14.3)	17.5 (11.2)	
Total PA (MET.h/week)	175.1 (84.2)	195.8 (116.3)	
MVPA (MET.h/week)	58.2 (53.7) *	81.0 (77.6) *	
Sedentary time (MET.h/week)	13.5 (10.5)	13.3 (8.5)	
24–28 weeks of gestation	Vit D N = 70	Placebo N = 62	Adjusted mean differences Vit D vs. Placebo
Sugar drinks (n/week)	4.2 (4.4)	4.4 (8.2)	−0.6 (−2.9; 1.7)
Fiber (n/week)	31.7 (19.4)	35.4 (31.8)	−5.8 (−14.8; 3.3)
Protein (n/week)	7.5 (3.8)	8.6 (6.4)	−1.4 (−3.2; 0,4)
Fat (n/week)	6.0 (5.6)	6.5 (7.2)	−0.1 (−2.2; 2.1)
Carbohydrates (n/week)	32.4 (21.4)	36.6 (40.0)	−6.3 (−12.7; 1.7)
Portion size (n/week)	17.4 (12.4)	17.0 (21.1)	0 (−6.2; 6.1)
Total PA (MET.h/week)	163.6 (66.1)	172.5 (90.2)	−1.3 (−22.2; 19.6)
MVPA (MET.h/week)	58.0 (50.9)	68.6 (64.3)	−3.1 (−19.7; 13.4)
Sedentary time (MET.h/week)	11.5 (8.1)	11.8 (7.7)	2.3 (−0.4; 5.0)
35–37 weeks of gestation	Vit DN = 67	PlaceboN = 57	Adjusted mean differences Vit D vs. Placebo
Sugar drinks (n/week)	4.1 (4.9)	3.6 (4.0)	0.4 (−1.4; 2.1)
Fiber (n/week)	31.0 (25.5)	33.9 (20.1)	−4.3 (−12.6; 4.0)
Protein (n/week)	7.9 (6.5)	8.4 (5.6)	−0.8 (−3.0; 1.3)
Fat (n/week)	7.4 (8.8)	7.9 (8.6)	0.3 (−2.5; 3.0)
Carbohydrates (n/week)	30.1 (17.9)	33.9 (21.7)	−5.5 (−18.2; 5.6)
Portion size (n/week)	18.3 (12.7)	17.8 (16.7)	−0.2 (−5.5; 5.1)
Total PA (MET.h/week)	133.9 (54.5)	147.8 (78.3)	−7.0 (−28.1; 14.1)
MVPA (MET.h/week)	37.4 (30.6)	51.2 (48.8)	−8.3 (−20.9; 4.4)
Sedentary time (MET.h/week)	13.8 (9.2)	11.5 (7.5)	−0.4 (−2.9; 2.0)

Definition of calculation of food scores for each time point (*n*/week) = Frequency/days per week x portions per serving (i.e., portion is the amount of food that can fit into the palm of your hand or equals 200 mL of fluid); sugar drinks: sugar drinks total = fruit juice total + soft drink total; fiber: total fiber = vegetables total + fruit total + whole-grain bread total; protein: protein total = meat and eggs total + fish total; fat: fat total = high-fat milk total + cakes and muffins total + fast food total; carbohydrates: carbohydrates total = cakes and muffins total + fruit total + whole-grain bread total + fruit juice total + soft drink total + potatoes/pasta total; portion size: portion size total = cakes and muffins total + high-fat milk total + fruit juice total + soft drink total + potatoes/pasta total. PA = physical activity, MVPA= moderate to vigorous physical activity.

**Table 4 nutrients-14-03781-t004:** Fetal cord blood, birth and pregnancy outcomes in the two intervention groups and mean differences between intervention groups.

Birth and Fetal Cord Blood Outcomes	Vitamin D *n* = 75		Placebo *n* = 64		Mean Difference (95% CI) Vit.D vs. Placebo †	Adjusted Mean Difference (95% CI) Vit.D vs. Placebo ††
	Mean	SD	Mean	SD		
Gestational age at birth, weeks	39.6	1.6	39.8	1.9	−0.10 (−0.69; 0.49)	0.22 (−0.38; 0.82)
Birth weight, g	3461.2	419.7	3504.5	458.9	−38.4 (−183.6; 106.9)	−49.9 (−203.3; 103.5)
Height, cm	52.1	3.1	52.0	3.1	0.2 (−0.9; 1.3)	0.1 (−1.0; 1.1)
Triglyceride, mmol/L	0.50	0.26	0.49	0.46	0.01 (−0.16; 0.18)	0.07 (−0.09; 0.24
LDL-C, mmol/L	0.94	0.38	0.96	0.50	−0.02 (−0.23; 0.20)	−0.22 (−0.24; 0.20))
HDL-C, mmol/L	0.57	0.17	0.64	0.21	−0.06 (−0.15; 0.02)	−0.06 (−0.16; 0.04)
FFA, mmol/L	0.261	0.163	0.264	0.259	−0.006 (−0.107; 0.095	0.023 (−0.084; 0.130)
3BHB, mmol/L	0.26	0.24	0.207	0.24	0.053 (−0.068; 0.174)	0.046 (−0.079; 0.171)
Leptin, ng/dL	10.1	8.1	9.6	8.3	0.5 (−3.5; 4.6)	−2.3 (−6.1; 1.5)
25-OH-D2, nmol/L	1.3	3.2	1.8	3.7	−0.4 (−1.4; 2.3)	−0.8 (−2.7; 1.1)
25-OH-D3, nmol/L	74.2 ***	22.1	51.0 ***	21.1	23.0 (11.3; 34.7) ***	22.3 (10.8; 33.8) ***
25-OH-D total, nmol/L	75.5 ***	21.9	52.8 ***	20.8	22.6 (11.0; 34.2) ***	21.3 (10.1; 32.6) ***
Skinfold triceps, mm	5.0	1.0	5.0	1.4	0.0 (−0.5; 0.4)	0.0 (−0.6; 0.5)
Skinfold subscapular, mm	5.1	1.4	4.9	1.1	0.2 (−0.3; 0.7)	0.2 (−0.4; 0.7)
Skinfold supra-iliac, mm	5.1	1.5	4.6	1.1	0.5 (−0.1; 1.0)	0.5 (−0.1; 1.1)
Skinfold thigh, mm	6.5	1.4	6.4	1.7	0.1 (−0.5; 0.7)	0.1 (−0.6; 0.8)
Neonatal sum of skinfolds, mm	21.8	4.4	21.0	4.3	0.7 (−1.0; 2.5)	0.9 (−1.0; 2.8)
Abdominal circumference, cm	33.4	2. 5	33.4	2.0	0.1 (−0.8; 1.0)	0.1 (−0.9; 1.1)
Head circumference, cm	34.8	1.4	35.1	1.7	−0.2 (−0.8; 0.3)	−0.1 (−0.7; 0.5)
	N	%	N	%	**OR (95%CI)**	**OR (95%CI)**
Male sex, *n*, %	33/74	44.6	33/64	51.6	0.78 (0.40; 1.53)	0.67 (0.32; 1.39)
LGA, *n*, %	7/73	9.6	9/60	15.0	1.57 (0.55; 4.54)	1.65 (0.51; 5.36)
Birth weight above 4000 g, *n*, %	7/73	9.6	9/60	15.0	1.57 (0.55; 4.54)	1.65 (0.51; 5.36)
SGA, *n*, %	0/75	0	2/61	3.3	N/A	N/A
Birth weight below 2500 g, *n*, %	0/75	0	1/64	1.6	N/A	N/A
Preeclampsia, *n*, %	1/66	1.5	3/58	5.2	4.00 (0.36; 44.30)	4.85 (0.26; 89.67)
Preterm birth, *n*, %	4/75	5.3	2/63	3.2	0.12 (0.01; 1.07)	0.13 (0.01; 1.20)
C-section, *n*, %	36/75	48.0	28/64	43.8	0.75 (0.37; 1.49)	0.56 (0.26; 1.25)
NICU, *n*, %	7/71	9.9	1/59	1.7	0.12 (0.01; 0.82)	0.13 (0.01; 1.20)

*** *p* < 0.001, HDL-C = high density lipoprotein cholesterol, LDL-C = low density lipoprotein cholesterol, FFA = free fatty acids, 3BHB = 3 hydroxybutyrate, 25-OH-D = 25 hydroxyvitamin, LGA = large for gestational age, SGA = small for gestational age, C-section = caesarean section, NICU = neonatal intensive care unit, † adjusted for gestational week except gestational week itself and fetal sex, †† adjusted for gestational week except gestational week itself, fetal sex, maternal age, maternal BMI at baseline, maternal HOMA Index at baseline, weight gain in pregnancy, study site, maternal baseline outcome variable if applicable, all results shown without correction for multiple comparisons.

## Data Availability

Data supporting reported results can be requested from the sponsor.

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
