# Peer review of "Vitamin D3 Supplementation in Overweight/Obese Pregnant Women: No Effects on the Maternal or Fetal Lipid Profile and Body Fat Distribution—A Secondary Analysis of the Multicentric, Randomized, Controlled Vitamin D and Lifestyle for Gestational Diabetes Prevention Trial (DALI)"

_nutrients, 2022, doi:10.3390/nu14183781_

Round 1
Reviewer 1 Report
The key finding is that vitamin D supplementation significantly increases a pregnant woman’s vitamin D levels but has no positive effect on her lipid profile (triglycerides, LDL-C, HDL-C, FFA, 3BHB or leptin). Nor are there positive effects on her baby in terms of gestational age at delivery, birth weight, height, skinfolds, and head or abdominal circumference. Vitamin D supplementation seems to be unnecessary for women during pregnancy.
Of course, 75% of the placebo group had sufficient vitamin D levels at the end of pregnancy. Would it be possible to compare them with the 25% who did not have sufficient levels? Did those two sub-groups differ in terms of lipid profile and birth outcomes?
Title
The title should mention the negative finding: “Vitamin D3 supplementation in overweight/obese pregnant women: no effect on the lipid profile and body fat distribution of the mother or the fetus…”
Differences in % European descent
The major flaw of this study is the failure to control for ethnic background:
· 91.1% of the treated group was of European descent.
· 78.7% of the placebo group was of European descent (p. 7).
That is a substantial difference. Although this difference is acknowledged on page 6, it is not discussed elsewhere in the paper as a possible limitation.
Vitamin D levels are lower in non-European populations, sometimes much lower. Hagenau et al. (2009) concluded that 25(OH)D levels are significantly higher in people of European origin than in those of non-European origin. Moreover, the latter have consistently low levels regardless of latitude. Those levels are low even when exposure to solar UV is frequent and intense. In south India, at 13.4°N, Harinarayan et al. (2007) found that 44% of the men and 70% of the women had 25(OH)D levels below 50 nmol/L. They were "agricultural workers starting their day at 0800 and working outdoors until 1700 with their face, chest, back, legs, arms, and forearms exposed to sunlight" [38]. That finding is corroborated by another Indian study, which found levels ≥ 50 nmol/L in only 31.5% of the participants, who had nonetheless been exposed to the sun for 5 hours every day (Goswami et al. 2008).
Vitamin D metabolism varies among human populations, as do many other aspects of physiology. Yet that factor was not controlled in this study. At the very least, the authors should investigate whether their results change significantly if they confine their analysis to the European participants.
References
Goswami, R.; Kochupillai, N.; Gupta, N.; Goswami, D.; Singh, N.; Dudha, A. Presence of 25(OH) D deficiency in a rural North Indian village despite abundant sunshine. Journal of the Association of Physicians of India. 2008, 56, 755-757.
Hagenau, T.; Vest, R.; Gissel, T.N.; Poulsen, C.S.; Erlandsen, M.; Mosekilde, L.; Vestergaard, P. Global vitamin D levels in relation to age, gender, skin pigmentation and latitude: an ecologic meta-regression analysis. Osteoporosis Int. 2009, 20, 133-140. https://doi.org/10.1007/s00198-008-0626-y
Harinarayan, C.V.; Ramalakshmi, T.; Prasad, U.V.; Sudhakar, D.; Srinivasarao, P.V.L.N.; Sarma, K.V.S.; Kumar, E.G.T. High prevalence of low dietary calcium, high phytate consumption, and vitamin D deficiency in healthy south Indians. Am. J. Clin. Nutr. 2007, 85, 1062-1067. https://doi.org/10.1093/ajcn/85.4.1062
Author Response
Dear Reviewer 1:
Thank you for your quick and precise evaluation of our manuscript and highlighting important aspects improving the quality of this paper. Indeed ethnicity is an important factor, which was not controlled for.
Title
The title should mention the negative finding: “Vitamin D3 supplementation in overweight/obese pregnant women: no effect on the lipid profile and body fat distribution of the mother or the fetus…”
A: We changed the title following your suggestion
Vitamin D sufficiency vs. insufficiency
Of course, 75% of the placebo group had sufficient vitamin D levels at the end of pregnancy. Would it be possible to compare them with the 25% who did not have sufficient levels? Did those two sub-groups differ in terms of lipid profile and birth outcomes?
A: We performed an analysis of vitamin D sufficient vs insufficient group at baseline and throughout pregnancy and analysed differences in birth outcomes. At baseline, no differences in lipid parameters or skinfolds were found (see table 1 in the word file below). After adjustment for baseline levels of the outcome variable, we did not find any differences at 24-28 weeks of pregnancy in lipid levels. At 35-37 weeks of gestation, significant lower levels of LDL-C were found in the group with sufficient vitamin D levels compared with insufficient vitamin D levels. However, we did not apply correction for multiple testing here and also need to keep in mind that numbers of pregnant women in the Vitamin D < 50nmol/l arm are small, especially at 35-38 weeks of pregnancy (n=106 vs. 12). Other parameters or lipid parameters did not differ between groups. Also birth outcomes and fetal lipid parameters were not different between sufficient and insufficient groups (table 2 in the word file below). Due to the low numbers (especially at 35-37 weeks of pregnancy and cord blood parameters) we decided against presenting this analysis in the manuscript, but of course advocate further research in this area.
Differences in % European descent
The major flaw of this study is the failure to control for ethnic background:
- 91.1% of the treated group was of European descent.
- 78.7% of the placebo group was of European descent (p. 7).
That is a substantial difference. Although this difference is acknowledged on page 6, it is not discussed elsewhere in the paper as a possible limitation.
Vitamin D levels are lower in non-European populations, sometimes much lower. Hagenau et al. (2009) concluded that 25(OH)D levels are significantly higher in people of European origin than in those of non-European origin. Moreover, the latter have consistently low levels regardless of latitude. Those levels are low even when exposure to solar UV is frequent and intense. In south India, at 13.4°N, Harinarayan et al. (2007) found that 44% of the men and 70% of the women had 25(OH)D levels below 50 nmol/L. They were "agricultural workers starting their day at 0800 and working outdoors until 1700 with their face, chest, back, legs, arms, and forearms exposed to sunlight" [38]. That finding is corroborated by another Indian study, which found levels ≥ 50 nmol/L in only 31.5% of the participants, who had nonetheless been exposed to the sun for 5 hours every day (Goswami et al. 2008).
Vitamin D metabolism varies among human populations, as do many other aspects of physiology. Yet that factor was not controlled in this study. At the very least, the authors should investigate whether their results change significantly if they confine their analysis to the European participants.
A: Thank you for this important comment and interesting references. Below you find tables showing number and distribution of ethnicities in the vitamin D trial (tables 3 and 4 in the word file below)
Adjusting for ethnicity (+baseline values) revealed comparable results in maternal and fetal lipid parameters (table 5 + table 6 in the word file below). We added this information in the manuscript. (p.9+ p.10) and also added a remark in the limitations section (p.13)
Analysis confined for European population as suggested by reviewer 1 did not reveal any differences in results or changed results between total population and European population only (table 7 + table 8 in the word file below). This analysis also demonstrates that ethnicity in this specific trial does influence the results significantly.
We conclude from our analysis that although differences in the distribution of ethnicity in the vitamin D and placebo arm of this study are present, this does not have significant effects on the results. Unfortunately, the population in this study is not big enough to allow further investigation for ethnical differences, but this is a very interesting topic and should be followed up in further studies.

Reviewer 2 Report
The authors conducted a study titled "Effects of Vitamin D3 supplementation in overweight/obese pregnant women on maternal and fetal lipids and body fat distribution – a secondary analysis of the multicentric, randomized, controlled Vitamin D And Lifestyle for gestational diabetes prevention" The manuscript is generally well-organized and within the scope of the journal. The results are trustworthy and can assist in enhancing the health of pregnant women. To improve the quality of the study, the following suggestions can be made to the authors:
The introduction is relatively lacking in strength. Please provide additional information on some pertinent studies in the same subject and remark on the significance of the work.
In the discussion section, it is recommended to provide the medical or health implications of your study's findings, which may help improve the health of patients or pregnant women.
Author Response
The introduction is relatively lacking in strength. Please provide additional information on some pertinent studies in the same subject and remark on the significance of the work.
A: Dear Reviewer 2:
Thank you for assessing our manuscript and giving guidance to improve the quality of this manuscript.
We tried to improve the introduction and provided additional information on some studies and a remark of significance.
In the discussion section, it is recommended to provide the medical or health implications of your study's findings, which may help improve the health of patients or pregnant women.
A: We added an additional remark providing medical/health implications of our findings in the discussion.